Report

# Olfactory sensory neurons transiently express multiple olfactory receptors during development

Longzhi Tan[1,†], Qian Li[2,†] & X Sunney Xie[1,*]

## Abstract

In mammals, each olfactory sensory neuron randomly expresses one, and only one, olfactory receptor (OR)—a phenomenon called the "one-neuron-one-receptor" rule. Although extensively studied, this rule was never proven for all ~1,000 OR genes in one cell at once, and little is known about its dynamics. Here, we directly tested this rule by single-cell transcriptomic sequencing of 178 cells from the main olfactory epithelium of adult and newborn mice. To our surprise, a subset of cells expressed multiple ORs. Most of these cells were developmentally immature. Our results illustrated how the "one-neuron-one-receptor" rule may have been established: At first, a single neuron temporarily expressed multiple ORs—seemingly violating the rule—and then all but one OR were eliminated. This work provided experimental evidence that epigenetic regulation in the olfactory system selects a single OR by suppressing a few transiently expressed ORs in a single cell during development.

**Keywords** immature and mature olfactory sensory neurons; odorant receptor; olfactory system; single-cell RNA-Seq; trace amine-associated receptor (TAAR)

**Subject Categories** Genome-Scale & Integrative Biology; Chromatin, Epigenetics, Genomics & Functional Genomics; Neuroscience

**Mol Syst Biol. (2015) 11: 844**

## Introduction

In mammalian olfactory systems, the ability to detect and discriminate between a tremendous number of odors relies on the "one-neuron-one-receptor" rule (Mombaerts, 2004). The main olfactory epithelium expresses the massive gene family of olfactory receptors (ORs)—G protein-coupled receptors that include more than 1,000 genes in the mouse genome (Buck & Axel, 1991). Despite the enormous family size of ORs, each olfactory sensory neuron is thought to randomly express one, and only one, OR—a phenomenon called the "one-neuron-one-receptor" rule (Serizawa et al, 2004). Neurons expressing the same OR coalesce into a specific set of glomeruli in the olfactory bulb of the brain (Mori & Sakano, 2011). The dual role

of ORs in odor detection and topographical mapping necessitates the "one-neuron-one-receptor" rule to ensure a precise translation of odor signals to the brain. However, the rule was only demonstrated by RNA *in situ* hybridization, genetic labeling, and single-cell RT–PCR (Malnic et al, 1999; Serizawa et al, 2003; Tietjen et al, 2003; Li et al, 2004; Shykind et al, 2004; Tian & Ma, 2008), none of which could probe all ~1,000 ORs at the same time. In addition, little is known about the dynamics of OR expression during development.

In this work, we present a direct test of the "one-neuron-one-receptor" rule by transcriptomic sequencing of single cells from adult and newborn mice. Our results provide experimental support for a previously proposed, yet not widely accepted, hypothesis that each cell may first transiently express multiple ORs and then eliminates all but one during development (Mombaerts, 2004). This is in sharp contrast to the popular view that only one OR is expressed at any given time, either through an irreversible choice (Li et al, 2004) or after switching between a few ORs (Shykind, 2005; Dalton et al, 2013). Such transient co-expression may provide a molecular basis for a recently discovered critical period of olfactory axon wiring in newborn mice (Ma et al, 2014; Tsai & Barnea, 2014).

## Results

From the main olfactory epithelium of mice, we sequenced 178 single cells with an average of 2.82 million single-end 100-bp or paired-end 50-bp reads per cell (standard deviation = 0.83 million, min = 1.06 million, max = 4.52 million; Fig 1A). The cells came from either adult mice, aged 1–3 months (56 cells), or newborn mice, on postnatal days 4–10 (122 cells) (Table EV1). In each cell, an average of 2,826 genes (standard deviation = 921, min = 805, max = 6,399) were detected above a threshold of 1 transcript per million (TPM) (Table EV1). The numbers of detected genes were similar between adult and newborn cells (median = 2,862 vs. 2,894, $P = 0.36$, two-sided Wilcoxon rank-sum test; Fig 1B). To ensure sample quality, after microfluidic capture, cells were stained for viability after capture and visually inspected to avoid multiple cells, and their cDNA size distributions were analyzed to exclude cells with RNA degradation.

We first established a time axis for neuronal development among the single cells. Because of continuous neurogenesis, the main

1   Department of Chemistry and Chemical Biology, Harvard University, Cambridge, MA, USA
2   Department of Cell Biology, Harvard Medical School, Boston, MA, USA
    *Corresponding author. Tel: +1 617 496 9925; E-mail: xie@chemistry.harvard.edu
    † These authors contributed equally to this work

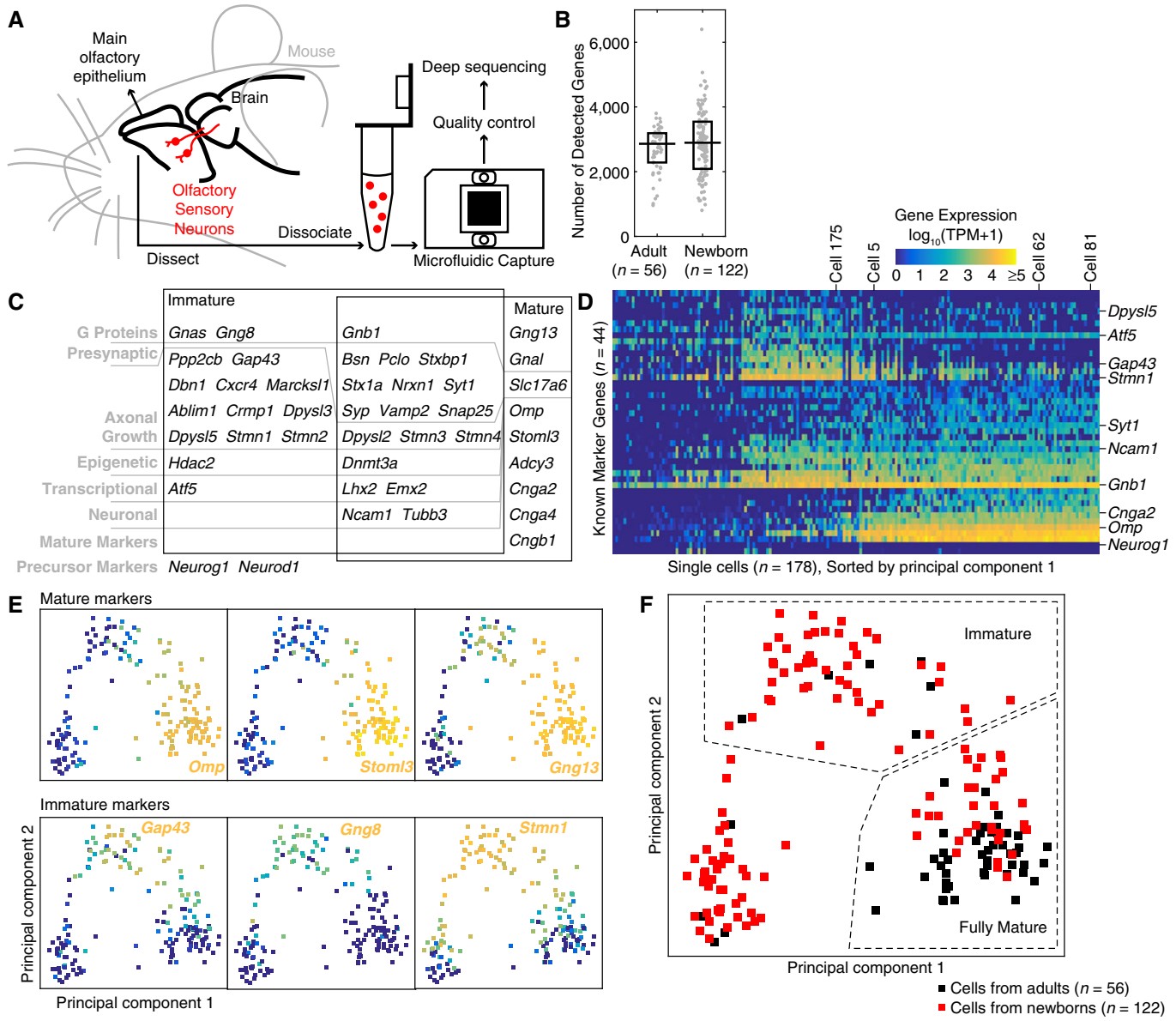

**Figure 1.  Single-cell transcriptomic sequencing of mouse olfactory sensory neurons established a time axis for neuronal development.**

A   We sequenced 178 single cells from the main olfactory epithelium of adult and newborn mice, with stringent quality control to avoid multiple cells, dead cells, or samples with RNA degradation.

B   Distribution of the number of detected genes among single cells. Similar numbers of genes were detected above a threshold of 1 transcript per million (TPM) in cells from adults and cells from newborns ($P = 0.36$, two-sided Wilcoxon rank-sum test). Each gray dot denotes one single cell. The horizontal line denotes the median, and the box denotes the lower and upper quartiles.

C   A total of 44 known marker genes were used in principal component analysis to infer the developmental stage of each single cell. These genes included most of the known markers for immature olfactory sensory neurons. Two black boxes indicate published expression patterns.

D   The 178 single cells can be roughly divided into 3 subpopulations along principal component 1. Genes were sorted first by published expression patterns (from top to bottom: immature only, both immature and mature, mature only, and precursor only) and then by average expression across all cells. Some representative genes and cells were labeled on the right and top of the panel. Color legend is above the panel.

E   Expression profiles for 3 example mature markers (*Omp*, *Stoml3*, and *Gng13*) and 3 example immature markers (*Gap43*, *Gng8*, and *Stmn1*) on principal component plots. Profiles for all 44 markers are shown in Appendix Fig S9.

F   Each single cell can be classified according to principal components 1 and 2.

olfactory epithelium contains a mixture of neurons at different developmental stages, from basally located immature neurons to apically located mature neurons (Verhaagen *et al*, 1989). To overcome the intrinsic stochasticity of gene expression and the technical noise and biases of single-cell RNA amplification (Wu *et al*, 2014), we combined 44 marker genes in principal component analysis to infer the developmental stages of single cells (Fig 1C). These genes were chosen to include most of the known markers for immature

olfactory sensory neurons (Calof & Chikaraishi, 1989; Belluscio *et al*, 1998; Roskams *et al*, 1998; Hirota & Mombaerts, 2004; Nedelec *et al*, 2004; MacDonald *et al*, 2005; Marcucci *et al*, 2009; McIntyre *et al*, 2010; Dalton *et al*, 2013; Sathyanesan *et al*, 2013), for mature olfactory sensory neurons (Monti-Graziadei *et al*, 1977; Belluscio *et al*, 1998; Bonigk *et al*, 1999; Kobayakawa *et al*, 2002; Zou *et al*, 2007; Sathyanesan *et al*, 2013), and for neuronal precursors (Cau *et al*, 1997; Table EV2). As expected, along principal component 1, these genes roughly divided the 178 single cells into 3 subpopulations: fully mature neurons, immature neurons, and other cells (including neuronal precursors, stem cells, and various types of supporting cells) from right to left (Fig 1D).

Principal components 1 and 2 together visualized the 3 subpopulations, with more mature neurons at the bottom right and more immature neurons on the top (Fig 1E). Note that the transition between immature and mature neurons is continuous, as shown by a considerable overlap in gene expression between classic immature markers *Gap43*, *Gng8*, *Stmn1* and classic mature markers *Omp*, *Gng13*, *Cnga2*, *Stoml3*, *Gnal* (Fig 1E and Appendix Fig S1). Changing the exact division line between immature and mature neurons did not affect our conclusions. Under our scheme of classification (Fig 1F), 54 immature neurons (6 from adults, 48 from newborns) were characterized by frequent expression of *Gap43*, *Gng8*, *Gnas*, *Dpsyl3*, *Dpsyl5*, *Hdac2*, relatively high expression of *Dpsyl2*, *Stmn1*, *Stmn2*, *Emx2*, *Lhx2*, *Tubb3*, and frequent absence of *Stmn4*, *Cnga4*, *Cngb1*, *Adcy3*. In contrast, 79 fully mature neurons (46 from adults and 33 from newborns) showed the opposite characteristics and were mostly positive for mature markers. The ratio between immature and fully mature neurons is much higher in newborns (1.45:1, compared to 0.13:1 in adults, $P = 2.5 \times 10^{-8}$, two-sided Fisher's exact test), consistent with published results (Verhaagen *et al*, 1989) and our RNA *in situ* hybridization of *Omp* and *Gap43* (Appendix Fig S1).

Our classification of immature and mature neurons is robust against the choice of marker genes. Instead of the 44 known marker genes from the literature, we picked another set of genes in a less supervised manner. A recent study conducted RNA sequencing on two FACS-sorted samples: *Neurog1*[+] neuronal precursors (a stage earlier than *Gap43*[+] immature neurons) and *Omp*[+] mature neurons (Magklara *et al*, 2011). The dataset contained 27,389 genes. Among the 496 genes that were highly expressed (FPKM > 100) in at least one sample, we picked the top 100 genes that were enriched in neuronal precursors and the top 100 in mature neurons based on fold changes (Table EV3). This set of 200 genes reproduced our main conclusions (Appendix Fig S2).

In each cell, we evaluated the expression of each OR with stringent criteria (Materials and Methods). In total, we made 153 confident calls of receptor expression, including 2 trace amine-associated receptors (TAARs) (Liberles & Buck, 2006), in particular *Taar4* in Cell 76 and *Taar7e* in Cell 74, and 1 vomeronasal receptor (VR) (Dulac & Axel, 1995), in particular *Vmn1r37* in Cell 101 (Fig 2A). The splicing isoforms that we observed are highly consistent with a recently published assembly of OR and VR transcripts, which was based on RNA sequencing of whole tissues (Ibarra-Soria *et al*, 2014). Out of all 151 confident calls of OR and VR expression, 120 cases (79%) have all splicing isoforms agreeing with the published assembly (Ibarra-Soria *et al*, 2014), and 13 cases (9%) contain a mixture of novel and published isoforms (Fig 2A). In addition, our

single-cell results allowed us to show for the first time that multiple isoforms of the same receptor can coexist in one single cell (Fig 2A). Figure 2C shows the coverage and splicing profiles of two such examples, *Olfr1507* and *Olfr536*. We also found that the level of receptor expression can differ by more than 3 orders of magnitude between single cells, with a median of TPM = $9.15 \times 10^3$, corresponding to ~1% of the transcriptome, and a range of TPM from 42.1 to $1.46 \times 10^5$, corresponding to 0.0042% to 15% of the transcriptome (Fig 2B). However, in 5 pairs of single cells that happened to choose the same ORs, the levels of OR expression were very similar within each pair, differing by less than three-fold (TPM values are *Olfr1507*: 1.24 vs. $1.96 \times 10^4$; *Olfr536*: 1.83 vs. $2.54 \times 10^4$; *Olfr672*: 1.14 vs. $1.46 \times 10^5$; *Olfr77*: 8.56 vs. $9.05 \times 10^3$; *Olfr1348*: 2.84 vs. $6.73 \times 10^4$). Such consistency suggests that each OR may be tightly regulated according to its own "set point", consistent with a reported role of OR expression levels in axonal targeting (Feinstein *et al*, 2004). The very high coverage of some ORs allowed us to identify novel exons or novel combinations of known exons that were previously undetected in whole tissues (Fig 2D).

To our surprise, we observed 20 cells—5 from adults, 15 from newborns—that expressed multiple ORs, seemingly violating the "one-neuron-one-receptor" rule. In total, we determined the status of OR expression for 155 out of 178 (87%) single cells (Table EV4). In addition to the 20 multi-receptor neurons, we found 57 cells that express no receptors (6 from adults and 51 from newborns) and 78 cells that express a single receptor (38 from adults and 40 from newborns, including 1 VR cell and 2 TAAR cells) (Fig 3A). We observed a tendency for more multi-receptor neurons in newborns compared to adults (($27 \pm 6$)% vs. ($12 \pm 5$)%, among cells with a single or multiple receptors, with standard error), consistent with an RNA *in situ* study in the septal organ (Tian & Ma, 2008), but the difference is not significant ($P = 0.077$, two-sided Fisher's exact test; Fig 3A). Despite their different numbers of detected ORs, single- and multi-receptor neurons have similar numbers of detected genes (median = 3,075 vs. 3,077, $P = 0.86$, two-sided Wilcoxon rank-sum test) (Fig 3B).

Interestingly, multi-receptor neurons showed a lower level of total OR expression compared to their single-receptor counterparts (median TPM = $1.18 \times 10^4$ vs. $1.75 \times 10^4$, $P = 0.034$, two-sided Wilcoxon rank-sum test; Fig 3C), suggesting a possibly earlier stage of OR expression. Each multi-receptor neuron expressed an average of 2.9 ORs, with a median of 2 and a range from 2 to 9. Figure 3D shows the level of total OR expression and the contribution of each OR in each of these cells. Figure 3E shows 3 examples of coverage profiles in multi-receptor neurons, and Appendix Figs S3–S6 show all 20 cells. Curiously, in Cell 5, we observed the co-expression of two adjacent ORs, *Olfr1030* and *Olfr1031*, on the same strand of chromosome 2 (Fig 3E). The two ORs are not highly homologous to each other (their previous names, *MOR196-2* and *MOR200-1*, indicate different OR subfamilies); yet in some isoforms, they share a same upstream exon. However, it is unclear whether such local co-expression can occur for other ORs.

We hypothesize that olfactory sensory neurons may express multiple ORs at an early developmental stage, based on the observation that multi-receptor neurons expressed ORs at a lower level (Fig 3C). Indeed, we detected the expression of *Gap43*, a gene critical for axon path-finding (Strittmatter *et al*, 1995; Maier *et al*, 1999) and a marker for immature neurons (Verhaagen *et al*, 1989), in 18

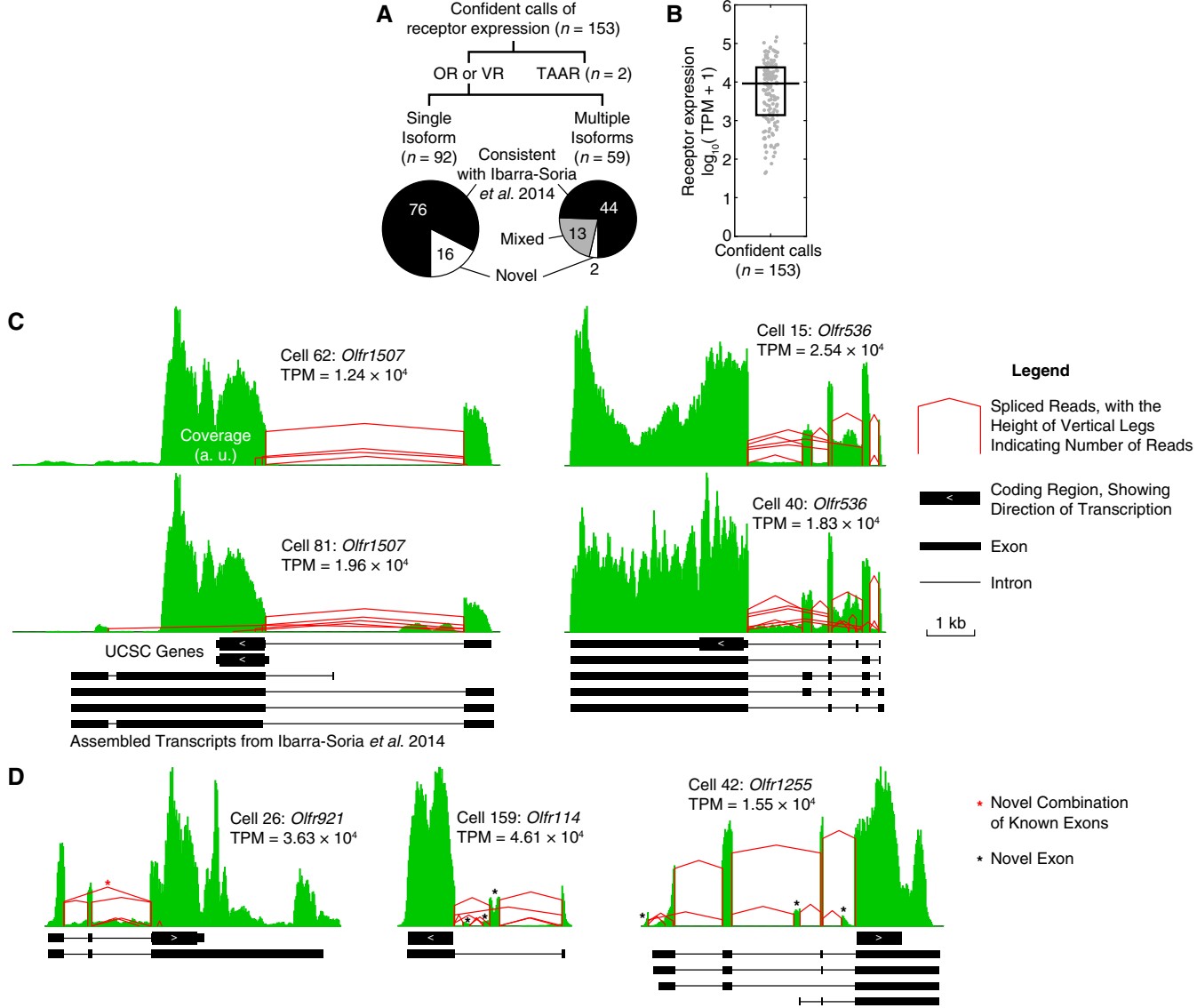

**Figure 2.  Expression of ORs was confidently detected in single cells.**

A    Composition of all confident calls of receptor expression. In each cell, we determined the expression status of each OR, VR, and TAAR and compared the transcripts with a recently published assembly (Ibarra-Soria *et al*, 2014).

B    Distribution of the level of receptor expression among all confident calls. The level of receptor expression can differ by more than 3 orders of magnitude between single cells. Each gray dot denotes a confident call of receptor expression. The horizontal line denotes the median, and the box denotes the lower and upper quartiles.

C    Two example ORs, *Olfr1507* and *Olfr536*, were expressed in multiple, co-existing isoforms. Note that some of *Olfr1507*'s transcripts do not contain the beginning of its coding sequence and are thus likely to be non-coding. Green pileups denote coverage by sequencing reads (arbitrary units, binned every 20 bp), and red lines denote spliced reads. Tracks below show annotated transcripts from UCSC genes (mm10) and from a recently published assembly of ORs and VRs (Ibarra-Soria *et al*, 2014). a. u., arbitrary units.

D    In 3 example ORs, *Olfr921*, *Olfr114*, and *Olfr1255*, we discovered novel exons (black asterisk) or exon combinations (red asterisk). Note that *Olfr1255*'s rightmost novel exon is likely part of a non-coding isoform. Symbols have the same meaning as in Fig 2C.

out of 20 multi-receptor neurons. Under our classification of immature and mature neurons (Fig 1F), 17 out of 20 (85%) multi-receptor neurons were immature (Fig 3F). This suggests that a substantial fraction of immature neurons—(57 ± 9)% among cells with a single or multiple receptors, with standard error—express more than one ORs, and this percentage drops dramatically to (4 ± 2)%, with standard error, when cells develop into fully mature neurons

($P = 1.8 \times 10^{-8}$, two-sided Fisher's exact test), restoring the "one-neuron-one-receptor" rule (Fig 3G).

An alternative explanation to our observations is that co-expressed ORs may arise from contamination from nearby cells or ambient RNA and that their enrichment in immature neurons may be an artifact because in fully mature neurons, the high level of existing OR expression may "mask" contamination and caused their

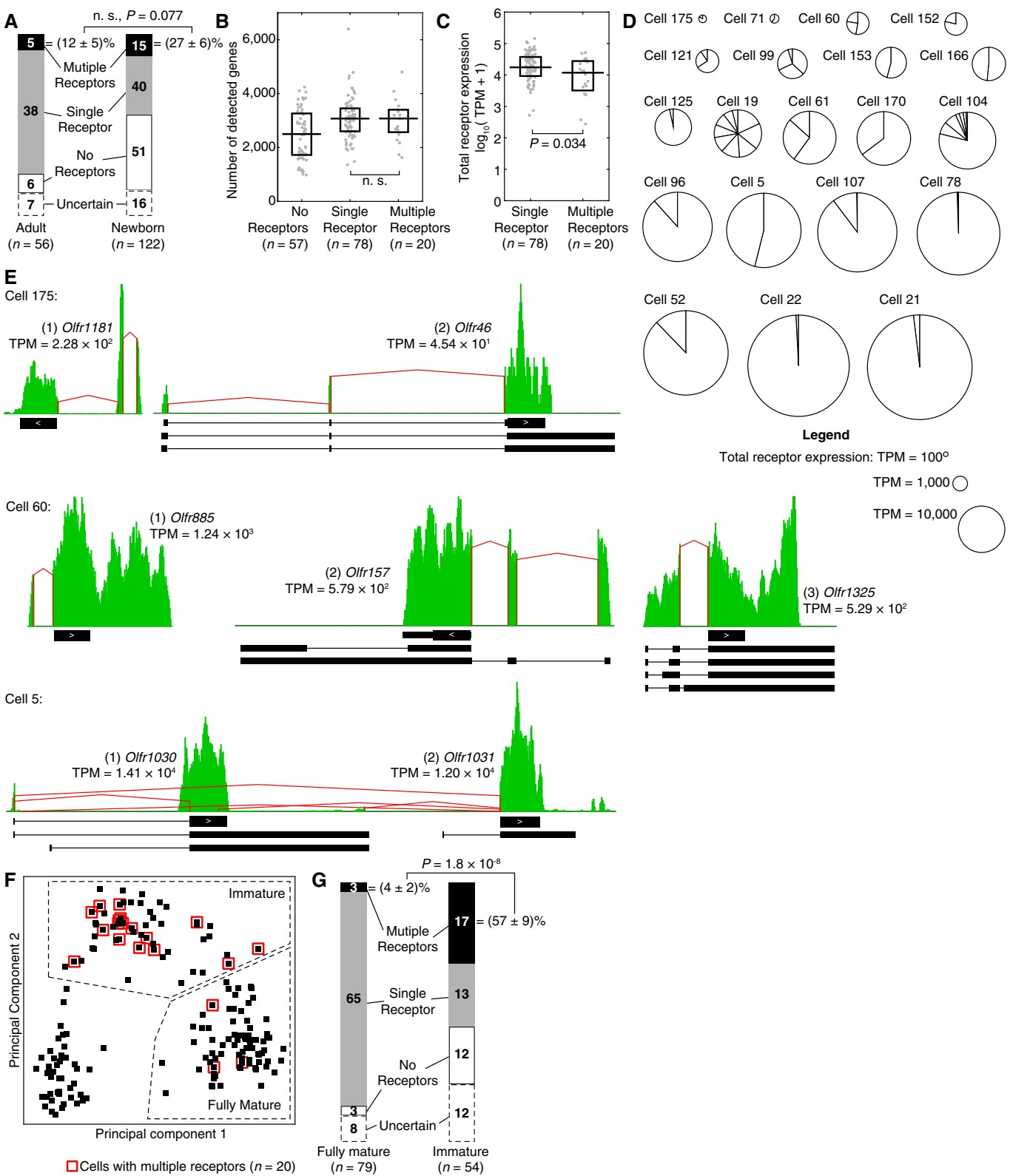

**Figure 3.**

apparent absence. Such "masking" may arise from competition for reverse transcription and/or PCR amplification. To rule out this possibility, we conducted a control experiment in which a "target" cell, expressing a single OR *Olfr1537* at TPM = $4.16 \times 10^{4}$, was either reverse-transcribed, amplified and sequenced alone, or processed as a 1:10 or 1:100 mixture with a "background" cell

**Figure 3.  A subset of 20 single cells expressed multiple ORs. Most of these cells were immature neurons.**

A   We determined the status of OR expression—expressing no receptors (white), a single receptor (gray), or multiple receptors (black)—for 155 out of 178 (87%) single cells. Among cells with a single or multiple receptors, newborn mice have a tendency to have more multi-receptor neurons, but the difference is not significant (two-sided Fisher's exact test). n. s., not significant.

B   Distribution of the number of detected genes among single cells with a single or multiple receptors. Similar numbers of genes were detected above a threshold of 1 TPM in single- and multi-receptor neurons ($P = 0.86$, two-sided Wilcoxon rank-sum test). Symbols have the same meanings as in Fig 1B. n. s., not significant.

C   Distribution of the total level of receptor expression among single cells. Multi-receptor neurons tend to have a slightly lower level of total OR expression compared to their single-receptor counterparts (two-sided Wilcoxon rank-sum test). Each gray dot denotes the sum of expression levels of all its ORs in a single cell. The horizontal line denotes the median, and the box denotes the lower and upper quartiles.

D   The level of total OR expression, denoted by the area of each circle, and the contribution of each OR, denoted by the size of each slice, in each of the 20 multi-receptor neurons.

E   Coverage profiles of 3 example multi-receptor neurons. Note that Cell 5 (bottom) expressed two adjacent ORs, *Olfr1030* and *Olfr1031*, and some isoforms share a same upstream exon. Symbols have the same meanings as in Fig 2C.

F   Same as Fig 1F, but with multi-receptor neurons labeled by red boxes.

G   A large fraction of immature neurons expressed multiple receptors, seemingly violating the "one-neuron-one-receptor" rule, while the rule was restored in fully mature neurons (two-sided Fisher's exact test).

(Appendix Fig S7A). Because the microfluidic device that was used for the main results does not support such operations, we conducted the control experiment with mouth pipetting (Li *et al*, 2013) and a similar chemistry (Picelli *et al*, 2014). In all 3 mixtures, we detected the "target" OR *Olfr1537* against the "background" of a cell that lowly expressed *Olfr728*, a cell that highly expressed *Olfr1348*, or a cell that expressed no receptors (Appendix Fig S7B). Therefore, a highly expressed OR does not seem to "mask" a co-expressed OR.

Single-cell transcriptomic sequencing is known to exhibit large measurement noise, especially for lowly expressed genes (Wu *et al*, 2014; Chapman *et al*, 2015), which may give rise to artifacts. To assess the extent of technical variations, we conducted an additional control experiment in which a single cell was split into two halves and processed separately (Appendix Fig S7C). The two halves showed great consistency quantifying highly expressed genes, such as the OR *Olfr107* and the markers *Omp*, *S100a5*, *Gng13*, *Gnal* (Appendix Fig S7D). They also agreed on the absence of genes such as *Gap43* and *Gnas*. However, for lowly or intermediately expressed genes, such as *Cnga2* in this cell, detection and/or quantification can be noisy. In particular, consistent detection between the two halves was frequent only for genes with TPM $> 10^3$, while "drop-outs" (detection in only one half) dominated for genes with TPM $< 10^2$ (Appendix Fig S7E). This suggests that in olfactory sensory neurons, genes with an expression level of TPM = $10^2$ to $10^3$, corresponding to 0.01–0.1% of the transcriptome, roughly constituted a minimal "unit" of reliable detection. In comparison, the expression levels of ORs in multi-receptor neurons were around or above this "unit". Therefore, our detection of multiple ORs in these cells was reliable.

The only remaining alternative explanation to our conclusions is that tissue dissociation may specifically damage immature neurons. For example, immature neurons may be more fragile than their mature counterparts and may thus take in more contaminating RNA; alternatively, immature neurons may be more prone to expressing multiple ORs in response to stress during dissociation. To conclusively rule out these possibilities, the same experiment needs to be reproduced in intact tissues, which is beyond the technical limits of current methods of multiplexed RNA *in situ* hybridization. As a first step toward such validation, we showed by two-color RNA *in situ* hybridization that at least in the TAAR olfactory subsystem (consisting of only 14 genes, in comparison with > 1,000 in the OR subsystem), olfactory sensory neurons indeed co-expressed two

receptors (*Taar6* and at least one of the five members in the *Taar7* family) at a frequency of ~10% in tissue cryosections of newborn mice (postnatal day 3; Appendix Fig S8A and B), which is much higher than published observations in adult animals (Liberles & Buck, 2006). Along the basal–apical axis, those cells were located in the middle of the olfactory epithelium (Appendix Fig S8C and D). This location indicates a transition from the immature *Gap43*$^+$ state to the mature *Omp*$^+$ state, which is consistent with our deep-sequencing results. The co-expression of *Taar6* and *Taar7* is not due to the presence of the pseudogene *Taar7c-ps*, because *Taar7c-ps*, as detected by a specific probe, was not co-expressed with *Taar6* (Appendix Fig S8E and F). Therefore, although conclusive validation would require a larger-scale experiment targeting all ~1,000 ORs with a high sensitivity, our conclusions can be partly validated in tissue cryosections.

## Discussion

For more than a decade, it has been extensively debated whether ORs are expressed one-at-a-time during the establishment of the "one-neuron-one-receptor" rule. We found that the popular view of one-at-a-time expression may not be true. The difference between our results and previous lineage-tracing experiments, in which OR choice seemed either permanent (Li *et al*, 2004) or mostly stable (Shykind *et al*, 2004), is likely due to the relatively low OR expression in multi-receptor neurons, which may be insufficient to drive robust *Cre* activity in a cell under translational arrest (Dalton *et al*, 2013) and with aggregated chromatin (Clowney *et al*, 2012).

Our findings suggest that epigenetic regulation behind the "one-neuron-one-receptor" rule is more complicated than previously thought, because current models cannot explain the elimination of all but one OR from multi-receptor neurons. One possible explanation to the transient nature of multi-receptor expression is a dramatic change in chromatin conformation during development, from a more permissive environment in immature neurons to a highly compacted one in fully mature neurons. This is consistent with recent genetic manipulations of several OR genes (Fleischmann *et al*, 2013). Although the same repressive histone mark H3K9me3 (Magklara *et al*, 2011) is present on OR genes throughout neuronal differentiation, our results suggest that additional epigenetic factors must be involved during the transition between immature and

mature neurons. Candidates include the gradual nuclear aggregation mediated by nuclear lamina and its receptor *Lbr* (Clowney *et al*, 2012), and the developmentally regulated subunits *Ezh2* and *Eed* of polycomb repressive complexes (PRCs) (Tietjen *et al*, 2003), which are known to compress chromatin with H3K27me3 (Armelin-Correa *et al*, 2014). Another possibility is that small-scale chromatin changes may block all but one ORs from enhancer networks that are crucial for gene expression (Markenscoff-Papadimitriou *et al*, 2014).

Our observed concurrence of multi-receptor expression, axonal growth, and synaptic formation leads to a speculation that OR regulation may be non-cell-autonomous. If so, the transient expression of multiple receptors may provide a molecular basis for a recently discovered critical period of olfactory axon wiring (Ma *et al*, 2014; Tsai & Barnea, 2014). Note that whatever the new mechanism is, the current kinetic model of epigenetic gene activation (Lyons *et al*, 2013; Tan *et al*, 2013) is still the biggest contributor to the "one-neuron-one-receptor" rule, bringing down the possible choices from more than 1,000 genes to < 10 in each single cell. Under this parameter regime, both activation and feedback can happen at the timescale of days, which is physiologically more feasible. Starting from there, receptor elimination may be the key to bringing the cell to the remarkable precision of "one-neuron-one-receptor".

Our current data have certain limitations. First, we avoided capturing multiple cells by visually inspecting the microfluidic device. However, it is possible that we missed very small pieces of contamination, which would introduce a small background rate of false positives. Such contamination would influence immature and mature neurons equally and thus would not affect our conclusions. Second, our data demonstrated the co-existence of mRNA molecules from different ORs in single cells. However, it remains unclear whether their genomic loci are simultaneously active. Future work on single-cell ChIP-Seq will help to distinguish between active co-transcription and residual transcripts from already silenced ORs, especially in cells with a dominant OR and one or more minor ORs (Fig 3D). Finally, we cannot rule out the possibility that multi-receptor neurons were later removed by cell death. If this is the case, multi-receptor expression will be permanent and lethal, rather than transient. However, this alternative scenario is unlikely because the majority of immature neurons need to be culled for the achievement of the "one-neuron-one-receptor" rule.

# Materials and Methods

## Single-cell sequencing of transcriptomes

All mouse experiments were performed in accordance with relevant guidelines and regulations. Animal protocols were approved by Harvard IACUC.

Mice came from the inbred strains C57BL/6J and C57BL/6NTac. Adult animals were 1–3 months old, and newborn animals were sacrificed on postnatal days 4–10.

The main olfactory epithelium was dissected and dissociated by the Papain Dissociation System (Worthington) at 37°C for 15 min and trituration for 5–15 times with a cut P1000 pipette tip. Cells were filtered by a 40-μm strainer (Falcon) and a 10-μm one (pluriSelect).

After spinning at 400 *g* for 2 min, cells were resuspended in DMEM (Gibco).

For the main experiments, cells were loaded at a concentration of ~750 K/ml onto a 5- to 10-μm mRNA-Seq C1 chip (Fluidigm). Cells were washed, stained by LIVE/DEAD Viability/Cytotoxicity Kit (Life Technologies), and discarded if stained red or if the chamber contained multiple cells. Amplified cDNA was harvested into 3 μl DNA dilution buffer (Fluidigm) per cell.

For the control experiments, cells were plated onto a cover glass coated with 10 ng/μl poly-D-lysine (Sigma). Cells were washed by HBSS (Gibco), picked by mouth pipetting, and amplified by Smart-Seq2 (Picelli *et al*, 2014) with minor modifications (22 cycles of PCR, with SuperScript II and its buffer replaced by ProtoScript II (NEB) to avoid bacterial contaminations in recent lots, and with two rounds of bead purification to minimize primer dimers).

Amplified cDNA was analyzed on high-sensitivity DNA chips (Agilent). Cells with a lot of short cDNA (< 1 kb) were discarded. Reads were aligned to the GRCm38/mm10 assembly of the mouse genome by TopHat 2.0.11 with default parameters. Transcript abundances were estimated by Cufflinks 2.2.1 with the annotation of UCSC genes and default parameters. Alignments were inspected in IGV (Broad Institute). TPM values were calculated after the removal of microRNAs, small nucleolar RNAs, and rRNAs from the Cufflinks output. Principal component analysis was done with the ranking of TPM values among all single cells.

## Evaluation of OR expression

In each cell, we carefully assessed the expression of each OR against three criteria: (i) Its coding sequence must be completely covered; otherwise, we may have detected a truncated, non-coding transcript; (ii) a large fraction of reads must have high mapping quality; otherwise, we may have detected mismapping from a homologous OR; (iii) some reads must span introns; otherwise, we may have detected contamination from genomic DNA. To minimize false negatives, the expression status of an OR is called "uncertain" when only 1 or 2 criteria are met.

In IGV, the distinction between a truly expressed OR and a mismapped one (namely, not satisfying criterion (b)) is very clear. A mismapped OR has a few narrow peaks in its coding region, where reads with low mapping quality (light colors in IGV) pile up, whereas the rest of the gene has no coverage. These peaks correspond to small stretches of nucleotide identity shared with a highly expressed, homologous OR in the same cell. In contrast, a truly expressed OR has a more uniform coverage across the coding region and UTRs. In most cases, especially in cells with single-end 100-bp reads, > 80% of all reads have high mapping quality. In only one case, this percentage dropped below 50%, probably because this particular OR was recently duplicated. Based on the coverage in its UTRs, we still qualified this OR.

## RNA *in situ* hybridization

*In situ* hybridization analysis of mouse main olfactory epithelium was performed as described before (Liberles & Buck, 2006). In Appendix Fig S1, cRNA riboprobes were used for *Omp* (992-base pair sequence amplified by primers CAAACGGCCAGCACTGATTC and ACCGGTACCACAGCCTATCT) labeled with fluorescein and

*Gap43* (907-base pair sequence amplified by primers AGATGGTGTC AAGCCGGAAG and CCGGGGTACAGTGCAAGAAT) labeled with digoxigenin. In Appendix Fig S8, cRNA riboprobes were used for *Taar7c-ps* (553-base pair sequence amplified by primers CAGAATA CCCAGATCTACTCTTGTC and CTTTAGGATTGTGACCATTCCTTT) labeled with digoxigenin and *Taar6* or *Taar7* (Liberles & Buck, 2006) labeled with fluorescein. Fluorescent images were taken on a Leica TCS SP5 II confocal microscope.

## Data availability

Raw sequencing data were deposited at the National Center for Biotechnology Information with accession number SRP065920 at the following link: http://www.ncbi.nlm.nih.gov/sra/SRP065920.

**Expanded View** for this article is available online.

## Acknowledgements

The authors thank Catherine Dulac, Chenghang Zong, Yirong Peng, and Alec R. Chapman for helpful discussion, Simone Picelli for answering questions about Smart-Seq2, and Jun Yong for early trials of dissection and sequencing. This work was supported by an NIH Transformative Research Award (R01 EB010244) and NIH Director's Pioneer Award (DP1 CA186693) to X.S.X.. L.T. was supported by an HHMI International Student Research Fellowship. Q.L. was supported by an NIH grant (R01 DC013289) to Stephen Liberles. We thank the Nikon Imaging Center at Harvard Medical School.

## Author contributions

LT, QL, and XSX designed the experiments. LT and QL carried out experiments and analyzed data. LT, QL, and XSX wrote the paper.

## Conflict of interest

The authors declare that they have no conflict of interest.

## Note added in proof

A report describing similar findings (Hanchate *et al*, Science 2015, DOI: 10.1126/science.aad2456) was published while our study was under consideration.

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
