## [Review Process File · Molecular Systems Biology]

Olfactory Sensory Neurons Transiently Express Multiple Olfactory Receptors during Development

Longzhi Tan, Qian Li and X. Sunney Xie

Corresponding author: X. Sunney Xie, Harvard University

Review timeline:

Submission date:	15 October 2015
Editorial Decision:	06 November 2015
Revision received:	08 November 2015
Accepted:	11 November 2015

Editor: Maria Polychronidou

Transaction Report:

1st Editorial Decision

06 November 2015

Thank you for submitting your work to Molecular Systems Biology. We have now heard back from the three referees who agreed to evaluate your manuscript. Overall, the reviewers acknowledge that the presented findings are a valuable contribution to the field. However, they raise a number of concerns, which should be carefully addressed in a revision of the manuscript.

The recommendations of the referees are rather clear, so there is no need to repeat the points listed below. Two rather fundamental issues raised by the reviewers are that further evidence needs to be provided to i) conclusively prove the existence of neurons expressing more than one ORs and ii) demonstrate the transient nature of multiple OR expression. While these additional analyses would significantly enhance the impact of the study (and therefore we would welcome their inclusion if available), we do not think that they are mandatory for the acceptance of this work. Nevertheless, the text should be carefully revised to avoid overstatements regarding these two points and to mention potential alternative explanations.

We have circulated the reports to all three reviewers as part of our 'pre-decision cross-commenting' policy. During this process, Reviewer #1, made the following comments (pasted below for your reference) regarding the concern of Reviewer #2 on the novelty of the study in view of the work by Tian et al, 2008: "I don't think the Tian et al, study should be considered to remove the novelty of this work. Tian et al, studied the septal organ, which is not the same as the main neuroepithelium, and expresses far fewer ORs. Thus it isn't unreasonable to think that coexpression could occur there, while not occurring in the main neuroepithelium. Their findings do anticipate the findings in this manuscript, but it cannot be said that the new findings were already established fact."

REFEREE COMMENTS

Reviewer #1:

Tan et al. describe single-cell transcriptomes, obtained by RNA-seq, of olfactory epithelium sensory neurons. The study addresses the one-neuron-one-olfactory-receptor rule, whereby it is thought that olfactory receptors are mutually exclusively expressed. By using single-cell RNA-seq, the authors were able to determine the full complement of olfactory receptors expressed in each single neuron, something which is very difficult to do by any other current method.

The main finding of the paper is the discovery that some neurons express multiple ORs simultaneously, and that those neurons belong to a less mature neuronal subclass. This is intriguing because it immediately suggests a mechanism of initial narrowing-down to a handful of ORs, followed by pruning to a single OR during neuronal maturation.

Overall I found the study compelling, well written and with clear figures. The single-cell RNA-seq was well performed and adequately controlled. The age of the animals was not mentioned, but should be.

My major concern is the issue of possible cross contamination. The authors ruled out obvious doublets by imaging the microfluidics chips, but it can still not be ruled out that a small piece of one neuron got stuck to another, which would be undetectable by imaging. Ideally, the findings should be validated by an independent method such as RNA FISH to unambiguously demonstrate the existence of double-labelled neurons. However, given that there are 1000+ ORs, I understand that it may be a nearly insurmountable challenge to find a neuron expressing a particular combination of ORs.

The fact that neurons with multiple ORs tended to be immature suggests that the finding could not be due to random cross-contamination, but this depends crucially on whether immature neurons tend to be neighbours or not. This could easily be determined from immunofluorescence imaging using markers of mature and immature neurons.

Given that conclusive proof of the existence of double-expressing neurons is lacking, I advise that these uncertainties should be clearly spelled out. At the same time, I think the data is sufficiently compelling and on the balance the finding is likely to be true. Therefore I recommend publication.

Reviewer #2:

The study by Tan/Li et al. used single-cell transcriptomics to study olfactory receptor expression (OR) in the olfactory epithelium of adult and newborn mice. They find that immature neurons express multiple ORs, and suggest that multi-OR expression could reflect a transient phase in development during which the locus experiences epigenetic reconstruction before only one OR becomes selectively expressed.

The major weakness of the manuscript is that the first conclusion is known (that immature neurons often express more than one OR) and the second conclusion regarding the transient expression dynamics is not fully supported by the data. Instead, the data presented in the manuscript is a systematic transcriptome-wide confirmation of a previous result of multi-OR expression in immature neurons.

Major points

1. The main concern with the manuscript is that the current data does actually not imply transient expression of multiple olfactory receptors (ORs), since other models (equally likely) were not ruled out. This is a major concern since the transient multi-OR-expression is the most novel claim by the authors.

"Transient" implies that something occurs over time followed by a change in state. However, in the manuscript there are no observations of cells with expression of multiple ORs that switch to express

a single OR. Instead the authors compared OR expression in immature and mature neurons, and found that multi-OR expression was more prevalent in immature neurons (both in adult and newborn mice). There could be multiple explanations for this observation. Immature cells that fail to express a single OR could be removed through processes such as cell death and apoptosis. In conclusion, the authors have not shown that olfactory neurons transiently express multiple OR during development and then reduce their OR-expression repertoire.

2. Novelty

The expression of ORs in immature and mature neurons has been investigated in earlier studies arriving at the same conclusions. The novelty here is the use of a more systematic transcriptome-wide method. For example Tian et al. 2008 (referenced in the manuscript) wrote: "The results suggest that there is a reduction of the sensory neurons expressing multiple ORs during postnatal development." Thus, previous studies have proposed that multi-OR expressing cells are more prevalent in immature neurons and less so in fully mature neurons, raising doubts to the novelty of that conclusions of the present manuscript.

3. Methods.

The method section is severely underexplained to the point that would not be possible to repeat the work nor to interpret the criteria for OR expression. The Methods section "Evaluation of OR expression" is particularly important for the interpretation of the study and must be improved. For example, these sentences "(b) a large fraction of reads must have high mapping quality, otherwise we may have detected mismatching from a homologous OR" is very unspecific and relates to the classification of OR expression in the cells, with direct consequences on the main conclusions (the fractions of cells expressing ORs in newborn/adult cells and immature-/mature-classified cells). For example, many of the disqualified OR-expressing cells in Table S2 have very high expression levels, but were disqualified (we assume due to criteria (b) under Evaluation of OR expression), and are marked "Yes" in the column "Coverage Profile Indicative of Mismatching". Exactly why were they disqualified? Why are all the other columns, related to intron-spanning reads etc, left blank in these cells? There is nowhere any information on how these calls were made.

Minor points

4. Cross-cell contamination.

The authors perform dilution experiments to estimate background signals (cell-to-cell contamination). Another simple approach to assess the possibility of cell-to-cell contamination could be to also look for marker expression of non-olfactory cells within olfactory cell libraries. Did the authors check for (for example) oligodentocyte, astrocyte marker expression within olfactory sequence libraries? Could these data be included as a supplementary item to support the lack of cell-to-cell contamination?

5.

Typo, page 4: "This suggests that a substantial(ly) fraction of immature neurons"

6.

Discussion, page 5: "For more than a decade, ORs have been thought to express one-at-a-time during the establishment of the "one-neuron-one-receptor" rule."

The sentence, as currently formulated, might give the impression that the authors discovered the exception to the one-neuron-one-receptor rules. Instead, reduction of multi-OR-expressing cells was actually observed in previous studies almost a decade ago (for example: Tian & Ma Mol Cell Neurosci. 2008 which is cited by the authors in the Results section). We therefore suggest this sentence to be modified.

7.

Unnecessary speculation. Discussion, page 5: "The transient nature of multi-receptor expression suggests a dramatic change in chromatin conformation during development, from a more permissive environment in immature neurons to a highly compacted one in fully mature neurons." First, the authors have to prove transient multi-OR expression (i.e. reversal of multi-OR expression in cells) - see major point 1. Second, there are no data on chromatin conformation in the paper, so this discussion is pure speculations. Third, why would it suggest a "drastic" change in chromatin conformation? - Could not OR-gene silencing equally well occur due to "smaller" chromatin

changes, like for example blocked enhancer interaction or any other "smaller" modification? We find all the comments on epigenetic silencing, drastic chromatin conformations to be unsupported and to carry little value.

Reviewer #3:

The study by Tan et al. constitutes a tour de force single cell analysis of the transcriptome of mouse olfactory neurons. This study differs from many emerging single cell analyses in that it is conducted in a very interesting system with extreme abundance of transcriptional variability, the olfactory epithelium, which is characterized by the unusual monogenic and monoallelic olfactory receptor (OR) gene expression. Previous work by many different groups has concluded that OR gene expression is likely singular, at least in mature, fully differentiated olfactory neurons. However, here, with the robustness of the single cell RNA-seq analysis that the Xie lab has pioneered, the authors reveal a somewhat different picture that suggests that before terminal differentiation olfactory neurons may co-express multiple OR genes before they culminate to the expression of a single OR allele. This finding is well supported by the data and is very important for the field, thus I recommend the publication of this paper. There are a few issues that need clarification:

1. In cells where the authors detect OR coexpression, are the co-expressed ORs from the same expression zone? Are the promoters of co-expressed ORs more similar than OR promoters that are never co-expressed? An effort to answer either of these questions would provide insight in the process of OR gene choice.
2. The authors find significant variability at the levels of OR mRNA- is this variability observed for the same OR gene among different neurons? If the authors have more than one cell that expresses the same OR in singular fashion, it would be important to know, since it has been proposed that OR expression levels are crucial for proper axonal targeting.
3. From browsing through the supplemental data it appears that some OR pseudogenes, for example *Olfrl372-ps1*, are expressed at high frequencies. Is this statistically significant? If yes the authors should talk about this, since pseudogenes are not able to elicit feedback.
4. Finally, an important issue that emerges from these findings is the issue of mRNA levels. Lineage tracing experiments (whereby Cre recombinase is co-expressed with an OR protein and permanently marks the cell that expressed that OR) and two-color ISH experiments in the OE have failed to detect the high frequency OR co-expression reported here. If indeed an immature olfactory neuron would express 2-9 OR genes before settling down to a single allele, then there should be a 2-9 fold more reporter positive neurons than the neurons that express the Cre-driving OR in lineage tracing experiments. However, this is not what is observed experimentally. This immediately suggests that either the mRNA levels of the co-expressed ORs are extremely low, or that these transcripts are not translated. If the former is true, then the levels of co-expressed ORs (at cells that express more than one OR) should be much lower than gene that have been previously used as effective Cre drivers in the OE (for example *Foxg1*, *Goofy*, *OMP*, *Lgr5*, *Krt14*, *Mash1*). In contrast, if the levels are relatively high, then the authors should be able to detect OR co-expression by two color RNA FISH, in immature olfactory neurons. Either scenario is very interesting,

In addition to these issues, there are some edits that I would like to propose and which would make the manuscript more accurate and more interesting for the field:

In the introduction, the authors reference two papers (Shykind et al., and Dalton et al.) as studies conflicting with their findings. These papers actually show frequent switching occurring in neurons that initially choose a functional OR (MOR28, and P2 in follow up paper by the Shykind group), which is consistent with the findings in this paper. The actually study that refutes these findings is Li et al., 2004, which shows that neurons choosing OR M71 never switch to a different allele. The authors should reference that paper and provide explanations for the differences between their findings and the results from Li et al.

In addition, in figure 3D, cells 96,107, 78, 52, 22 and 21 seem to have one dominant OR and one that is significantly less expressed. The authors interpret this result as evidence for co-expression but I would argue that the same result would be obtained if the neuron is currently expressing the

dominant OR, while it was previously expressing the weakly detected OR, from which it had switched off. In other words, the authors should entertain the interpretation that they detect the products of OR gene switching instead of OR coexpression, which would be very plausible if OR mRNA is not unusually unstable. This scenario is also consistent with the fact that multiple ORs are detected more frequently in immature neurons. In follow up studies, the authors could test this by combining single cell RNAseq with lineage tracing analysis. Regarding the other cells shown in Figure 3D, it is very interesting that these are lowly expressing cells with multiple ORs. Previous work revealed molecular events contributing to singular OR expression, thus I am wondering if the authors could provide some additional information on these cells (175-104) that would provide insight to the mechanisms of singular OR expression. Specifically, it would be interesting to know if these cells have detectable levels of *Lsd1* and *Lbr* mRNA. Other groups have shown that developmentally regulated downregulation of *Lbr* is necessary for the aggregation of OR loci and that ectopic expression of *Lbr* leads to OR coexpression at low levels- thus it would be interesting to know if these multi-OR expressing cells are still *Lbr* positive. Similarly, *Lsd1* downregulation seems important for stable OR expression, thus, it would be interesting to know if these cells are *Lsd1* positive.

1st Revision - authors' response

08 November 2015

Reviewer #1:

Tan et al. describe single-cell transcriptomes, obtained by RNA-seq, of olfactory epithelium sensory neurons. The study addresses the one-neuron-one-olfactory-receptor rule, whereby it is thought that olfactory receptors are mutually exclusively expressed. By using single-cell RNA-seq, the authors were able to determine the full complement of olfactory receptors expressed in each single neuron, something which is very difficult to do by any other current method.

The main finding of the paper is the discovery that some neurons express multiple ORs simultaneously, and that those neurons belong to a less mature neuronal subclass. This is intriguing because it immediately suggests a mechanism of initial narrowing-down to a handful of ORs, followed by pruning to a single OR during neuronal maturation.

Overall I found the study compelling, well written and with clear figures. The single-cell RNA-seq was well performed and adequately controlled. The age of the animals was not mentioned, but should be.

In our initial submission, we described the age of the animals on Page 2, in the first paragraph of Result: “The cells came from either adult mice, aged 1 to 3 months (56 cells), or newborn mice, postnatal day 4 to 10 (122 cells)”. In the current revision, we clarified it again in the second paragraph of Materials and Methods to make this information more visible.

My major concern is the issue of possible cross contamination. The authors ruled out obvious doublets by imaging the microfluidics chips, but it can still not be ruled out that a small piece of one neuron got stuck to another, which would be undetectable by imaging. Ideally, the findings should be validated by an independent method such as RNA FISH to unambiguously demonstrate the existence of double-labelled neurons. However, given that there are 1000+ ORs, I understand that it may be a nearly insurmountable challenge to find a neuron expressing a particular combination of ORs.

We agree with the reviewer that a small piece of contamination may be undetectable by imaging. We clarified this on Page 6, in the last paragraph of Discussion.

The fact that neurons with multiple ORs tended to be immature suggests that the finding could not be due to random cross-contamination, but this depends crucially on whether immature neurons tend to be neighbours or not. This could easily be determined from immunofluorescence imaging using markers of mature and immature neurons.

This is a good point. In our initial submission, we showed by two-color RNA FISH (currently Appendix Figure S1, which is also consistent with previous studies) that immature neurons cluster together in basal layers, whereas mature ones cluster together in apical layers. Morphologically, immature and mature neurons have similar sizes of cell bodies, and similar distances between neighbors. This supports our conclusions because if contamination happened, it would affect immature and mature neurons equally.

Given that conclusive proof of the existence of double-expressing neurons is lacking, I advise that these uncertainties should be clearly spelled out. At the same time, I think the data is sufficiently compelling and on the balance the finding is likely to be true. Therefore I recommend publication.

This is a good point. Please refer to our summary of major changes.

Reviewer #2:

The study by Tan/Li et al. used single-cell transcriptomics to study olfactory receptor expression (OR) in the olfactory epithelium of adult and newborn mice. They find that immature neurons express multiple ORs, and suggest that multi-OR expression could reflect a transient phase in development during which the locus experiences epigenetic reconstruction before only one OR becomes selectively expressed.

The major weakness of the manuscript is that the first conclusion is known (that immature neurons often express more than one OR) and the second conclusion regarding the transient expression dynamics is not fully supported by the data. Instead, the data presented in the manuscript is a systematic transcriptome-wide confirmation of a previous result of multi-OR expression in immature neurons.

We agree with the reviewer that the theory of multi-receptor expression is not novel. Throughout our initial submission, we made it very clear that our major contribution is not the proposal of the theory but its experimental validation. For example, we stated “Our results provide experimental support for a previously proposed, yet not widely accepted, hypothesis that each cell may first transiently express multiple ORs, and then eliminates all but one during development” on Page 1, in the second paragraph of Introduction. Our revised Discussion further clarified this issue.

Major points

1. The main concern with the manuscript is that the current data does actually not imply transient expression of multiple olfactory receptors (ORs), since other models (equally likely) were not ruled out. This is a major concern since the transient multi-OR-expression is the most novel claim by the authors.

"Transient" implies that something occurs over time followed by a change in state. However, in the manuscript there are no observations of cells with expression of multiple ORs that switch to express a single OR. Instead the authors compared OR expression in immature and mature neurons, and found that multi-OR expression was more prevalent in immature neurons (both in adult and newborn mice). There could be multiple explanations for this observation. Immature cells that fail to express a single OR could be removed through processes such as cell death and apoptosis. In conclusion, the authors have not shown that olfactory neurons transiently express multiple OR during development and then reduce their OR-expression repertoire.

The reviewer raised a valid concern about potential cell death of these immature, multi-receptor neurons. We believe that this alternative scenario is unlikely, because the observed fraction of multi-receptor neurons in immature neurons (57%) would require the majority of immature neurons to be culled for the achievement of the “one-neuron-one-receptor” rule. Nonetheless, we added discussion of this alternative scenario on Page 6, to the last paragraph of Discussion.

2. Novelty

The expression of ORs in immature and mature neurons has been investigated in earlier studies arriving at the same conclusions. The novelty here is the use of a more systematic transcriptome-wide method. For example Tian et al. 2008 (referenced in the manuscript) wrote: "The results suggest that there is a reduction of the sensory neurons expressing multiple ORs during postnatal

development." Thus, previous studies have proposed that multi-OR expressing cells are more prevalent in immature neurons and less so in fully mature neurons, raising doubts to the novelty of that conclusions of the present manuscript.

We agree with the pre-decision cross-comment on this issue of novelty (see above, editor's comments). In addition, we want to point out that in contrast to our manuscript, Tian *et al.* (2008) did not distinguish between immature and mature neurons (on Page 485, "The OR positive neurons at P0 contained both mature (OMP-positive) and immature OSNs (OMP-negative) ... we observed that 2.0% of the OSNs (both mature and immature) expressing more than one receptor ... The results suggest that there is a reduction of the sensory neurons expressing multiple ORs during postnatal development"). In fact, they did not positively stain for immature-only markers (such as *Gap43*). Therefore, our conclusions are not the same as Tian *et al.* (2008).

3. Methods.

The method section is severely underexplained to the point that would not be possible to repeat the work nor to interpret the criteria for OR expression. The Methods section "Evaluation of OR expression" is particularly important for the interpretation of the study and must be improved. For example, these sentences "(b) a large fraction of reads must have high mapping quality, otherwise we may have detected mismapping from a homologous OR" is very unspecific and relates to the classification of OR expression in the cells, with direct consequences on the main conclusions (the fractions of cells expressing ORs in newborn/adult cells and immature-/mature-classified cells). For example, many of the disqualified OR-expressing cells in Table S2 have very high expression levels, but were disqualified (we assume due to criteria (b) under Evaluation of OR expression), and are marked "Yes" in the column "Coverage Profile Indicative of Mismapping". Exactly why were they disqualified? Why are all the other columns, related to intron-spanning reads etc, left blank in these cells? There is nowhere any information on how these calls were made.

Thank you for this point.

The single-cell sequencing part of this manuscript was carried out according to the standard procedure of a Fluidigm C1 machine. With a detailed C1 manual, one can easily reproduce our experiments.

Regarding the evaluation of OR expression, these ORs were disqualified because of mismapping, most likely from homologous regions of a dominant OR in the same cell. Although they may seem highly expressed, the TPM values are always much lower than the dominant OR where mismapping originated. Therefore, they can be safely rejected. To avoid confusion, we added a paragraph to Materials and Methods, elaborating on this issue. We also clarified the meaning of blank Excel entries (representing "No") in the legend of Table EV4.

Minor points

4. Cross-cell contamination.

The authors perform dilution experiments to estimate background signals (cell-to-cell contamination). Another simple approach to assess the possibility of cell-to-cell contamination could be to also look for marker expression of non-olfactory cells within olfactory cell libraries. Did the authors check for (for example) oligodentocyte, astrocyte marker expression within olfactory sequence libraries? Could these data be included as a supplementary item to support the lack of cell-to-cell contamination?

The reviewer proposed to check cell-to-cell contamination by inspecting genes that are not supposed to express in immature and mature neurons. However, the transcriptomes of immature neurons and of non-neuronal populations in the olfactory epithelium (for example, sustentacular cells, basal cells, immune cells, and olfactory ensheathing glia) remain relatively poorly characterized.

On the other hand, despite the diversity of non-neuronal cells, even in newborn mice *Omp*⁺ mature neurons are already the dominant population, as shown by our RNA *in situ* hybridization on postnatal day 3 (Appendix Figure S2). Therefore, *Omp*⁺ mature neurons are the primary source of potential contamination. The lack of mature markers in the multi-receptor neurons is therefore a good indicator of the lack of contamination.

5.

Typo, page 4: "This suggests that a substantial(ly) fraction of immature neurons"

Thank you for pointing this out. We have corrected this typo.

6.

Discussion, page 5: "For more than a decade, ORs have been thought to express one-at-a-time during the establishment of the "one-neuron-one-receptor" rule."

The sentence, as currently formulated, might give the impression that the authors discovered the exception to the one-neuron-one-receptor rules. Instead, reduction of multi-OR-expressing cells was actually observed in previous studies almost a decade ago (for example: Tian & Ma Mol Cell Neurosci. 2008 which is cited by the authors in the Results section). We therefore suggest this sentence to be modified.

We changed this sentence into "For more than a decade, it has been extensively debated whether ORs are express one-at-a-time during the establishment of the "one-neuron-one-receptor" rule."

7.

Unnecessary speculation. Discussion, page 5: "The transient nature of multi-receptor expression suggests a dramatic change in chromatin conformation during development, from a more permissive environment in immature neurons to a highly compacted one in fully mature neurons." First, the authors have to prove transient multi-OR expression (i.e. reversal of multi-OR expression in cells) - see major point 1. Second, there are no data on chromatin conformation in the paper, so this discussion is pure speculations. Third, why would it suggest a "drastic" change in chromatin conformation? - Could not OR-gene silencing equally well occur due to "smaller" chromatin changes, like for example blocked enhancer interaction or any other "smaller" modification? We find all the comments on epigenetic silencing, drastic chromatin conformations to be unsupported and to carry little value.

The reviewer raised a valid concern about our proposed mechanism for the elimination of all but one ORs. We clarified this by changing the sentence in question into "One possible explanation to the transient nature of multi-receptor expression is ..." and by adding the reviewer's alternative mechanism "Another possibility is that small-scale chromatin changes may block all but one ORs from enhancer networks that are crucial for gene expression (Markenscoff-Papadimitriou *et al.* 2014)" to the same paragraph. We believe that given such an intriguing phenomenon, some mechanistic speculation is helpful for the understanding of the olfactory system and is thus suitable for a short paragraph in Discussion.

Reviewer #3:

The study by Tan et al. constitutes a tour de force single cell analysis of the transcriptome of mouse olfactory neurons. This study differs from many emerging single cell analyses in that it is conducted in a very interesting system with extreme abundance of transcriptional variability, the olfactory epithelium, which is characterized by the unusual monogenic and monoallelic olfactory receptor (OR) gene expression. Previous work by many different groups has concluded that OR gene expression is likely singular, at least in mature, fully differentiated olfactory neurons. However, here, with the robustness of the single cell RNA-seq analysis that the Xie lab has pioneered, the authors reveal a somewhat different picture that suggests that before terminal differentiation olfactory neurons may co-express multiple OR genes before they culminate to the expression of a single OR allele. This finding is well supported by the data and is very important for the field, thus I recommend the publication of this paper. There are a few issues that need clarification:

1. In cells where the authors detect OR coexpression, are the co-expressed ORs from the same expression zone? Are the promoters of co-expressed ORs more similar than OR promoters that are never co-expressed? An effort to answer either of these questions would provide insight in the process of OR gene choice.

This is a good point. Regarding the zones, we looked at the zonal information for all 20 multi-receptor neurons. However, most of the ORs that we detected do not have published zonal

information. Therefore, our current data cannot rule out the possibility that ORs from different zones may be co-expressed.

Regarding the promoters, we obtained the DNA sequences directly upstream our detected transcript start sites of these co-expressed ORs. Unfortunately, initial inspection did not find any common elements. This is consistent with published studies, which described difficulties in finding common elements in Class I OR promoters, or in OR promoters with known zonal information.

2. The authors find significant variability at the levels of OR mRNA- is this variability observed for the same OR gene among different neurons? If the authors have more than one cell that expresses the same OR in singular fashion, it would be important to know, since it has been proposed that OR expression levels are crucial for proper axonal targeting.

Thank you for pointing this out. We do have 5 pairs of single-receptor cells expressing the same ORs, and the expression levels are indeed similar within each pair. We added this new information on Page 3, to the middle paragraph of Results.

3. From browsing through the supplemental data it appears that some OR pseudogenes, for example Olfr1372-ps1, are expressed at high frequencies. Is this statistically significant? If yes the authors should talk about this, since pseudogenes are not able to elicit feedback.

Olfr1372-ps1 is likely an artifact caused by sequencing reads originating from genomic DNA. We and others have noticed that single-cell sequencing of olfactory sensory neurons tends to generate many reads from genomic DNA, even though the protocol was designed not to lyse the nucleus. After sequencing many single cells, we found these genomic reads to be mostly reproducible, and part of *Olfr1372-ps1* usually has high occurrence of these reads.

We agree that pseudogenes are an interesting topic; but given their poor annotation in the mouse genome, we decided not to study them in this manuscript.

4. Finally, an important issue that emerges from these findings is the issue of mRNA levels. Lineage tracing experiments (whereby Cre recombinase is co-expressed with an OR protein and permanently marks the cell that expressed that OR) and two-color ISH experiments in the OE have failed to detect the high frequency OR co-expression reported here. If indeed an immature olfactory neuron would express 2-9 OR genes before settling down to a single allele, then there should be a 2-9 fold more reporter positive neurons than the neurons that express the Cre-driving OR in lineage tracing experiments. However, this is not what is observed experimentally. This immediately suggests that either the mRNA levels of the co-expressed ORs are extremely low, or that these transcripts are not translated. If the former is true, then the levels of co-expressed ORs (at cells that express more than one OR) should be much lower than gene that have been previously used as effective Cre drivers in the OE (for example Foxg1, Goofy, OMP, Lgr5, Krt14, Mash1). In contrast, if the levels are relatively high, then the authors should be able to detect OR co-expression by two color RNA FISH, in immature olfactory neurons. Either scenario is very interesting,

This is a good point. We added detailed discussion on this issue to the first paragraph of Discussion.

In addition to these issues, there are some edits that I would like to propose and which would make the manuscript more accurate and more interesting for the field:

In the introduction, the authors reference two papers (Shykind et al., and Dalton et al.) as studies conflicting with their findings. These papers actually show frequent switching occurring in neurons that initially choose a functional OR (MOR28, and P2 in follow up paper by the Shykind group), which is consistent with the findings in this paper. The actually study that refutes these findings is Li et al., 2004, which shows that neurons choosing OR M71 never switch to a different allele. The authors should reference that paper and provide explanations for the differences between their findings and the results from Li et al.

This is a good point. On Page 2, in the last paragraph of Introduction, to the end of the sentence “This is in sharp contrast to the popular view that only one OR is expressed at any given time”, we added “either through an irreversible choice (Li et al. 2004) or after switching between several ORs

(Shykind 2005, Dalton, Lyons, and Lomvardas 2013).” In addition, we added more discussion on Page 5, to the first paragraph of Discussion.

In addition, in figure 3D, cells 96,107, 78, 52, 22 and 21 seem to have one dominant OR and one that is significantly less expressed. The authors interpret this result as evidence for co-expression but I would argue that the same result would be obtained if the neuron is currently expressing the dominant OR, while it was previously expressing the weakly detected OR, from which it had switched off. In other words, the authors should entertain the interpretation that they detect the products of OR gene switching instead of OR coexpression, which would be very plausible if OR mRNA is not unusually unstable. This scenario is also consistent with the fact that multiple ORs are detected more frequently in immature neurons. In follow up studies, the authors could test this by combining single cell RNAseq with lineage tracing analysis.

This is a good point. We clarified this issue of active co-transcription versus residual mRNA from an already silenced OR on Page 6, in the last paragraph of Discussion.

*Regarding the other cells shown in Figure 3D, it is very interesting that these are lowly expressing cells with multiple ORs. Previous work revealed molecular events contributing to singular OR expression, thus I am wondering if the authors could provide some additional information on these cells (175-104) that would provide insight to the mechanisms of singular OR expression. Specifically, it would be interesting to know if these cells have detectable levels of *Lsd1* and *Lbr* mRNA. Other groups have shown that developmentally regulated downregulation of *Lbr* is necessary for the aggregation of OR loci and that ectopic expression of *Lbr* leads to OR coexpression at low levels- thus it would be interesting to know if these multi-OR expressing cells are still *Lbr* positive. Similarly, *Lsd1* downregulation seems important for stable OR expression, thus, it would be interesting to know if these cells are *Lsd1* positive.*

The reviewer raised a good point about further characterizing gene expression in specific multi-receptor neurons. We have looked at candidate genes such as *Lsd1* and *Lbr*; but they were lowly expressed and thus poorly detected in our dataset. This is partly expected because, for example, our theoretical model (Tan *et al.* 2013) dictates that *Lsd1* must be very lowly expressed to ensure the “one-neuron-one-receptor” rule. Future work should isolate multi-receptor neurons in larger quantities to build up statistical power.